# A Novel Betulinic Acid Analogue: Synthesis, Solubility, Antitumor Activity and Pharmacokinetic Study in Rats

**DOI:** 10.3390/molecules28155715

**Published:** 2023-07-28

**Authors:** Yucen Liang, Meixuan Zhu, Tao Xu, Weimin Ding, Min Chen, Yang Wang, Jian Zheng

**Affiliations:** 1Key Laboratory of Saline-Alkali Vegetation Ecology Restoration in Oil Field, Ministry of Education, College of Life Sciences, Northeast Forestry University, Harbin 150040, China; liangyucen1119@163.com (Y.L.); ywang1971@hotmail.com (Y.W.); 2Changchun Institute of Biological Products Co., Ltd., Changchun 130011, China; 3School of Chemical and Environmental Engineering, Harbin University of Science and Technology, Harbin 150040, China

**Keywords:** betulinic acid analogue, equilibrium solubility, anti-tumor activity, mitochondrial pathway, pharmacokinetics

## Abstract

Betulinic acid (BA) and betulin (BE) are naturally pentacyclic triterpenes with documented biological activities, especially antitumor and anti-inflammatory activity. However, their bioavailability in vivo is not satisfactory in terms of medical applications. Thus, to improve the solubility and bioavailability so as to improve the efficacy, 28-O-succinyl betulin (SBE), a succinyl derivative of BE, was synthesized and its solubility, in vitro and in vivo anti-tumor activities, the apoptosis pathway as well as the pharmacokinetic properties were investigated. The results showed that SBE exhibited significantly higher solubility in most of the tested solvents, and showed a maximum solubility of 7.19 ± 0.66 g/L in *n*-butanol. In vitro and in vivo anti-tumor activity assays indicated both BA and SBE exhibited good anti-tumor activities, and SBE demonstrated better potential compared to BA. An increase in the ratio of Bad/Bcl-xL and activation of caspase 9 was found in SBE treated Hela cells, suggesting that the intrinsic mitochondrial pathway is involved in SBE induced apoptosis. Compared with BA, SBE showed much-improved absorption and bioavailability in pharmacokinetic studies.

## 1. Introduction

Triterpenes are an important class of natural bioactive products, which contain a 30-carbon skeleton as the main feature. Among these compounds, pentacyclic triterpenes, one of the most significant subclasses, are shown to possess notable medical properties including antitumor, anti-HIV infection and anti-inflammatory activity as well as hepatocyte protection effect [1,2]. Betulinic acid (BA, 3β-hydroxy-lup-20(29)-en-28-oic acid), a lupane-type pentacyclic triterpene, was found to exhibit a number of biological activities, particularly selective antitumor effect [3]. Studies showed that BA exerts its effects directly on the mitochondrion pathway and induces selective death of tumor cells, i.e., most of the healthy cells and normal tissues would not be affected by BA [4]; accordingly, it has very slight toxicity and relatively a higher safety in normal cells, even the dosage up to 500 mg/kg body weight [2]. Betulin (lup-20(29)-en-3β, 28-diol, BE, Figure 1), which is also known as betuline, betulinol or betulinic alcohol [5], was one of the first natural products isolated from plants in 1788 and many studies have demonstrated that BE possesses a broad range of biological and pharmacological properties [6]. The chemical structure of BE was determined in 1952, which is the 28-hydroxyl analog of BA (Figure 1) [7,8]. BE was commonly found in many plants including some fruits as well as some herbs, and especially, it is even presented in large amounts in the outer layer of birch bark (up to 30% dry weight) [6,8]. Therefore, BE is often used as a cheap starting material for the chemical preparation of BA and other related compounds. The semi-synthesis of BA derivatives from BE or BA has been widely reported, as well as their pharmacological bioactivities [8,9,10]. However, the use of these compounds as potential therapeutic agents is hindered due to their limited solubility as well as their low bioavailability in vivo [11]. Fortunately, the hydroxyl groups of BE/BA gave researchers the opportunity to convert them into more soluble derivatives using some chemistry approaches. To date, lots of reports have demonstrated their simple or advanced modifications of BE/BA including the in vitro cytotoxicity studies, however, there are not many studies involving in vivo activity and the pharmacokinetics characteristics [2,4,6,8,9,11,12].

In the present study, a strategy to keep the structural functional group and improve the solubility and bioavailability so as to improve the efficacy was proposed. A novel betulinic acid analog, 28-O-succinyl betulin (SBE) was semi-synthesized by introducing a succinyl group at the C-28 site of BE, and its solubility, in vitro and in vivo anti-tumor activities, the apoptosis pathway as well as the pharmacokinetic properties were investigated.

## 2. Results and Discussion

### 2.1. Chemistry

Both BA and BE are naturally bioactive compounds extensively spread throughout the plant kingdom. Unfortunately, they are practically insoluble, and overall absorption as well as the therapeutic effect is not satisfactory. Thus, lots of research was focused on structure modification to achieve better solubility and enhanced bioactivities, especially the simple modifications at the C-3, C-20, and C-28 sites (Figure 1). Bulky and electron-donating groups introduced at these sites of BE/BA would be favorable for their solubility and bioactivity. Since BE is the 28-hydroxyl analog of BA and shares the same pentacyclic triterpenoid core with BA, BE was often selected as the starting material for the modification as BE is extracted in higher amounts than BA quantitatively. In this case, a novel betulinic acid analog (SBE) was semi-synthesized by introducing a succinyl group at the C-28 site of BE. The introduction of the short chain might be better for lipophilicity, while the carboxyl group could improve the water solubility to a certain extent. In order to introduce an electronegative potential at the end of the side chain at the C-28 site, we employed succinic anhydride to form a carboxylic group at the end of the side chain through esterification of the C-28 hydroxyl group of BE (Figure 1).

### 2.2. Solubility and Apparent Oil/Water Partition Coefficient (P) of SBE

All measured solubility of SBE and BA in different solvents are shown in Table 1. Compared to BA, SBE showed significantly higher solubility in water, petroleum ether, acetonitrile, *n*-butanol and methanol. Among all the tested solvents, the solubility of SBE in *n*-butyl alcohol was found to have the best solubility (7.19 ± 0.66 g/L).

The distribution of drugs in membranes was related to their partition coefficients in bulk oil/water systems [13,14]. The partition coefficient of SBE in water-saturated *n*-octanol versus PBS at different pH (from 5–8, simulating the pH in the intestinal tract) was also obtained using an indirect method by determining its concentration in water-saturated *n*-octanol before and after distribution. The measured solubility and apparent oil/water partition coefficient of SBE in phosphate buffer at different pH are shown in Table 2. The solubility of SBE in PBS at pH 7–8 was significantly higher than that of SBE at pH 5–6.5, indicating SBE was much more hydrophilic in neutral and basic environments. Accordingly, the lipophilicity of SBE in PBS at pH 7–8 was also enhanced. All the Log*P* values were between 0.91 and 0.99, suggesting SBE is soluble in both hydrophilic and hydrophobic environments and might have good membrane permeability.

### 2.3. In Vitro and In Vivo Antitumor Activities

The in vitro cytotoxic potency of SBE was measured using an MTT assay against six different cancer cells including MPC2, HT29, DU145, NCI-H520, Hela and 2774 cell lines, with BA as the positive control. The effects at different concentrations (0–50 μM) of both BA and SBE were assessed (Relative growth rate of different cancer cells incubated with different concentration of compounds was shown in Appendix A). As shown in Table 3, SBE exhibited much higher inhibitory activity than BA against most of the test tumor cell lines except for DU145 and NCI-H520 with no significant difference. Meanwhile, SBE also demonstrated higher cytotoxicity than that of those BE derivatives with phthalyl, maleyl, and the hexahydrophthalyl groups at the same C-28 position of BE established in our previous study [8]. These results indicated that the introduction of succinyl moiety on BE might significantly increase the anti-tumor activity of BA analogs.

In order to determine the efficacy of SBE in vivo, the Lewis lung carcinoma (LLC) subcutaneous xenograft growth model was established in C57BL/6J mice. BA was used as the positive control. The mice in the test groups received BA or SBE (0.4 mmol/kg) via intraperitoneal injection every other day, while the placebo group received an equal amount of ethanol/tween 80/normal saline (1/1/18, *v/v/v*). In the placebo group, the mean tumor volume reached 3522.72 ± 2446.63 mm^3^ in 18 days, whereas that of the SBE- and BA-treated group was 398.17 ± 384.06 mm^3^ and 1340.91 ± 1186.99 mm^3^, respectively (Figure 2A). The tumor growth inhibition rate (IR) of SBE and BA was about 88.69% and 61.93%, respectively. The mean tumor weight of SBE- and BA-treated mice was 0.42 ± 0.40 g and 0.89 ± 0.53 g, respectively; While that of vehicle control was 2.86 ± 1.16 g (Figure 2B), and the tumor weight was reduced by about 85.31% and 68.89% by SBE and BA, respectively. The differences in tumor growth inhibition and tumor weight between the BA-treated mice and placebo controls were not statistically significant (*p* > 0.05). The differences between the SBE-treated mice and BA-treated mice were also not statistically significant (*p* > 0.05). However, there were significant differences observed between the SBE-treated group and placebo controls (*p* < 0.05) (refer to Figure 2A,B). These results suggest that SBE might be more potent than BA against the LLC model. Additionally, the pictures depicting the removal of tumors from tumor-bearing mice during the anatomical examination also demonstrated a relatively higher level of antitumor activity compared to BA. There was no obvious effect on the body weight of the mice for all three groups (Figure 2D) compared with the vehicle control group, indicating no toxicity of BA and SBE to mice.

### 2.4. Effects of SBE on the Expression of Cleaved-cas9, Bad and Bcl-xL

Lots of studies have reported that the mitochondria play a key role in the intrinsic pathway of mammalian cell apoptosis, and the mitochondrial permeability transition could directly trigger the apoptosis process. Early studies have demonstrated that a decrease in the mitochondrial inner transmembrane potential was found in the SHEP neuroblastoma cells treated with BA, suggesting that the intrinsic mitochondrial pathway might be involved in BA-induced apoptosis [15]. Our previous study also showed that BA activated the mitochondria pathway, which led to the apoptosis of HeLa cells [16]. In this case, we speculated that the apoptosis pathway induced by SBE might be also associated with the mitochondrial pathway.

According to mitochondrial outer membrane permeabilization assay during the mitochondria-dependent apoptosis stage, it is associated with most of the pro-apoptotic stimuli, and this process is controlled mostly by both pro- and anti-apoptotic members of the Bcl-2 family [17]. Among these protein members, Bad belongs to the pro-apoptotic proteins, while Bcl-xL belongs to the anti-apoptotic members. Thus, to test whether SBE induced cell death is mediated by mitochondrial pathways, the levels of Bad (belong to pro-apoptotic proteins) and Bcl-xL (belong to anti-apoptotic proteins) were determined in SBE treated Hela cells via western blot analysis. As shown in Figure 3, the expression level of Bad in SBE treated Hela cells was significantly higher than that in the control cells. Although there was no significant difference in Bcl-xL levels in SBE and BA treated Hela cells, the ratio of Bad/Bcl-xL was notably up-regulated in both BA and SBE treated Hela cells, suggesting that SBE/BA treatment facilitated the expression of Bad, and indicating the mitochondria pathway participated in the intrinsic apoptosis.

The important role of caspases during mitochondria-dependent apoptosis pathway was also widely recognized. Caspases belong to the family of cysteinyl aspartate specific proteases involved in the apoptosis process and mainly include the two groups of initiators (caspases 8, 9 and 10) and executioners (caspases 3, 6 and 7) [18]. To identify the effect of SBE on the mitochondrial apoptosis pathway, the level of caspase 9 was examined here. As shown in Figure 3, detectable cleavage products of caspase 9 clearly increased after treatment with SBE and BA, which indicates that caspase 9 was activated.

These results showed that mitochondrial perturbations upon treatment with SBE resulted in an increase in the ratio of Bad/Bcl-xL and activation of caspase 9, leading to cell death; thus, the apoptosis pathway induced by mitochondria was involved in the effects of the SBE treatment of HeLa cells, which was consistent with that in the BA treated Hela cells [16].

### 2.5. Pharmacokinetic Study

It is well known that the poor solubility of BA/BE results in a low effective concentration and limited absorption for medical efficacy, limiting its application and development as a pharmaceutical preparation. Thus, it is important to determine the pharmacokinetic properties of SBE. In our present study, the pharmacokinetic study was performed in rats after i.v. (5 mg/kg, *n* = 6) and oral (200 mg/kg, *n* = 6) administration of SBE.

The mean plasma concentration-time profiles were plotted in Figure 4, and the pharmacokinetic parameters are shown in Table 4. A non-compartmental pharmacokinetic analysis model was used here to calculate the pharmacokinetic parameters, and the results are shown in Table 4. After a single i.v. dose of 5 mg/kg, the level of SBE reached a relatively higher level immediately, and then declined in the following test hours (Figure 4A). The plasma concentration declined with T_1/2_ of 9.77 ± 2.70 h and MRT of 13.51 ± 3.74 h. The AUC_0–48 h_ and AUC_0-∞_ values of SBE were 2473.03 ± 706.60 and 2729.27 ± 776.23 h·ng/mL, respectively.

After oral dosing, SBE showed a slow oral absorption phase in rats (Figure 4B and Table 3) with a peak concentration of 1042.76 ± 259.11 ng/mL at about 4 h after administration, and then decreased in the following 24 to 48 h, which meaning that the absorption of SBE reached the maximum concentration within four hours, and then the drug elimination was dominant in the following test time. The MRT and T_1/2_ were 6.43 ± 1.64 and 11.13 ± 2.03 h, respectively. The AUC_0–48_ h and AUC_0−∞_ values of SBE obtained were 9385.70 ± 2902.72 and 9719.28 ± 2910.56 h·ng/mL, respectively. In our previous studies, the AUC_0−∞_ value of BE after an i.g. dose of 500 mg/kg was only 1408.3 ± 183.7 h·ng/mL [3], which was about 5% of that of SBE, indicating the astonishing enhanced absorption of SBE than that of BE. The oral bioavailability of SBE in the present study was 9.49%, calculated following the equation given in our previous literature [19].

## 3. Materials and Methods

BA and BE were purchased from Boyle Chemical Co, Ltd. (Shanghai, China). All other chemicals (analytically pure or HPLC grade) were commercially available.

### 3.1. Synthesis of SBE

Analytical thin layer chromatography (TLC) was carried out on precoated silica gel GF254 plates (E. Merck, Darmstadt, Germany). Yields refer to the isolated yield of the product after purification by silica-gel column chromatography (200–300 mesh). The melting point (m.p.) was measured on an X-6 micro melting-point apparatus (Beijing Tech Instrument Co., Beijing, China). Fourier Infrared spectroscopy (FTIR) analyses were performed using a Nicolet Mattson Infinity Gold FTIR spectrometer using KBr pellets. ^1^H nuclear magnetic resonance (^1^H-NMR) spectra (300 MHz) and ^13^C nuclear magnetic resonance (^13^C-NMR) spectra (75 MHz) were recorded in CDCl_3_ at 25 °C with tetramethylsilane (TMS) and solvent signals as reference on a Bruker AVANCE-300 MHz. Chemical shifts are reported in ppm (δ). High-resolution mass spectra (HRMS) were obtained using AB Sciex Triple TOF 5600 (AB Sciex, Redwood City, CA, USA). Elemental analyses were carried out using a Thermo Finnigan EA 1112 Series Flash Elemental Analyzer.

BE (2.5 mmol), succinic anhydride (2.6 mmol) and pyridine (5 mL) were heating refluxed in 50 mL of methylene dichloride under stirring in a round-bottom flask. The reaction was monitored by TLC. After BE disappeared on TLC, the reactant was thoroughly washed with saturated sodium bicarbonate and sodium chloride solutions, and the mixture was dried over anhydrous magnesium sulfate and subsequently filtered. After that, the solvent was removed under reduced pressure, and then the crude product was purified by column chromatography with a mobile phase of petroleum ether/ethyl acetate (*v/v* = 2/1). The desired product fractions were pooled and dried to yield 0.71 g (64.5%) white solid of SBE, m.p. 239–241 °C; 1H NMR (CDCl_3_, 300 MHz) was as follows: δ 4.69 (s, 1H, H-29_a_), 4.59 (s, 1H, H-29_b_), 4.32 (d, 1H, H-28_a_), 3.90 (d, 1H, H-28_b_), 3.22 (dd1H, H-3), 2.71–2.63 (dt, 4H, H-2′; H-3′), 2.48 (1H, H-19); ^13^C NMR (CDCl_3_, 75 MHz) was as follows: δ177.3 (C1′), 172.5 (C4′), 150.1 (C20), 109.9 (C29), 79.1 (C3), 63.2 (C28); IR (film) ν_max_: 3387, 2941, 2868, 1725, 1453, 1246, 1217, 882, 751 cm^−1^; molecular formula: C_34_H_54_O_5_; HRMS: *m/z* = 541.3812 (calculated 541.3893) [M-H]^−^; elemental analysis, observed: C, 75.05; H, 9.97 (Calculated: C, 75.23; H, 10.03). (The spectra were shown in Appendix A)

### 3.2. Determination of the Equilibrium Solubility and Apparent Oil/Water Partition Coefficient of SBE

The solubility was recorded at 20 °C in the present study. The solubility under equilibrium conditions was determined using the saturation shake-flask method with minor modifications [20]. Briefly, an excess amount of SBE was added slowly to 1 mL of different solvents including methanol, ethanol, ethyl acetate, *n*-butanol, *n*-octanol, acetonitrile, petroleum ether as well as water. The mixture was then in crimp-sealed vials and was stirred vigorously at 20 °C for about 48 h. After that, the mixture was left to stand for an additional 24 h to reach equilibration, and then followed by centrifuging at 13,000 rpm for at least 10 min. 50 µL of the supernatant was transferred into a new tube, and evaporated to dryness under a slow nitrogen stream. The residue was then stored at −20 °C for further use or reconstituted in 1 mL of methanol. For the aqueous solution, 500 µL of the supernatant was transferred to the new tube, and the residue was reconstituted in 100 µL of methanol after dryness. After being centrifuged at 13,000 rpm for 10 min, a 20 μL aliquot was injected into the HPLC systems.

Equilibrium solubility and apparent oil-water partition coefficient of SBE in pH 5.0, 6.0, 6.5, 7.0, 7.4 and 8.0 phosphate buffer were also determined. Equilibrium solubility was determined by the method described above. The apparent oil-water partition coefficient (P) was calculated through the mass concentration ratio of the oil phase and water phase after balancing SBE distribution using the following equations: *P* = C_1_V_1_/(C_0_V_0_ − C_1_V_1_), where C_0_ and C_1_ are the initial and final concentrations of SBE in *n*-octanol before and after equilibration, respectively; V_0_ is the volume of *n*-octanol (water-saturated), V_1_ is the final oil volume after equilibration.

### 3.3. In Vitro and In Vivo Antitumor Activities

The in vitro cytotoxic activities of SBE were evaluated on six different tumor cell lines including MPC2 (pancreatic cancer cell line), HT29 (colon cancer cell line), DU145 (prostate cancer cell line), NCI-H520 (lung cancer cell line), Hela (cervical cancer cell line) and 2774 (ovarian cancer cell line) using MTT assay [8,21]. Cells were cultured in RPMI DMEM medium (containing 100 IU/mL G-penicillin, 100 µg/mL streptomycin and supplemented with 10% fetal calf serum) at 37 °C with 5% of CO_2_ in a CO_2_ incubator. In the MTT experiments, the cells were initially seeded in a 96-well plate with 0.1–0.2 mL (about 15 × 10^3^ cells) of culture medium per well and allowed to attach overnight (about 24 h). Once attached, the cell confluence in each well reached approximately 50–60% when the drugs were added. Then the medium was replaced with fresh medium containing SBE or BA at different concentrations in four replicates. After incubating in the CO_2_ incubator for 3 days (about 72 h), 30 µL of MTT solution (3 mg/mL in culture medium) was added to each well and incubated for an additional 4 h. The medium was then aspirated and 150 µL of DMSO was added to each well to dissolve the formazan crystals thoroughly. The absorbance (OD value) was quantitated at 490 nm using a microplate reader (ELX800, Biotek, America). The relative growth inhibition rate (%) was calculated as (1 − mean absorbance of the sample/mean absorbance of the control) × 100%, considering the optical density of the control as 100%.

The antitumor activities in vivo assay were performed as described previously with minor modifications [12,22,23,24]. C57BL/6J mice (20 ± 2 g) for the xenograft animal model were purchased from the Laboratory Animal Center of Jilin University (Jilin, China). A formulation of SBE (48 mg/mL) or BA (40 mg/mL) was prepared by mixing SBE or BA with ethanol/tween 80/normal saline (*v/v/v* = 1/1/18). Each of the mice was inoculated subcutaneously on the back with about 1 × 10^6^ Lewis lung carcinoma (LLC) tumor cells (purchased from Nanjing KeyGen Biotech. Inc., China) to obtain the xenograft animal model. Then the mice were divided into three experimental groups (*n* = 4) randomly on the next day after implantation. BA (200 mg/kg, i.e., 0.4 mmol/kg), SBE (240 mg/kg, i.e., 0.4 mmol/kg) or placebo was injected intraperitoneally (i.p.) every other day. The changes in tumor length, width and body weight were also observed and recorded every other day. Xenograft tumor volume (TV) and growth inhibition rate (IR) were calculated as the following equation [25]:TV = 0.5 × length × width^2^;
IR = (V_control_ − V_treat_)/V_control_ × 100%.

Mice were continually observed for 18 days, and then sacrificed on day 18 via euthanasia. The implanted tumors then were excised and weighed for subsequent analysis. All the animal handling procedures above were according to standard operating procedures approved by the institutional animal care and use committee at Northeast Forestry University.

### 3.4. Western Blotting Analysis

Expression of the proteins including cleaved-cas9, Bad and Bcl-xL in Hela cells treated with BA or SBE were determined by western blot analysis. Briefly, HeLa cells (1.7 × 10^5^) were seeded in a 6-well microplate and allowed to attach overnight (about 24 h). The cells were then treated with BA or SBE at a concentration of 50 μmol/L for another 24 h. After that, cells were collected and lysed in PIRA buffer (Beyotime Biotech, Beijing, China) with PhosSTOP phosphatase inhibitor cocktail tablets (Roche Diagnostics, Mannheim, Baden-Württemberg, Germany). Bradford method was used to quantify the protein concentration [26]. The samples were denatured at 100 °C for 5 min using a block heater. Then 30 μg of the denatured samples were separated by SDS-PAGE and allowed to transfer to a PVDF membrane. After the blots were blocked and washed, they were incubated overnight at 4 °C with specific primary antibodies against cleaved-cas9, Bad, Bcl-xL and β-actin. The blotting membrane was then washed three times using TBS-tween (Tris buffer saline with 1% Tween-20, pH 7.5), and incubated with IgG secondary antibodies for another 1 h at room temperature (or incubated overnight at 4 °C), followed by the same washing procedures above. After that, the immunoreactive proteins were developed with the ECL system. Imagelab software was used to quantify proteins and β-actin was used as a loading control. The changed fold was represented by the protein expression ratio [(target protein/the β-actin)/(the control group/the β-actin)].

### 3.5. Pharmacokinetic Study

Male Wistar rats (200 ± 20 g) used in the experiments were obtained from the Laboratory Animal Center of Jilin University (Changchun, China). After arrival, rats were acclimated to a 12-h light/dark cycle in a temperature-controlled environment with appropriate humidity for at least one week before treatment. For the intragastric (i.g.) administration groups, rats needed to fast for 8–12 h (overnight) before treatment while water was freely available.

The rats were divided into two groups (*n* = 6) randomly, and all the rats were taken jugular vein cannulation [19,27] before intravenous (i.v.) administration with a single dose of 5 mg/kg SBE as well as intragastric (i.g.) administration with a single dose of 200 mg/kg SBE. A series of blood samples (150–200 μL) from the jugular vein were collected into sodium heparin-containing tubes before and at 0, 0.08, 0.17, 0.33, 0.5, 1.0, 2.0, 4.0, 6.0, 8.0, 12.0, 24.0, 36.0 and 48.0 h time points after administration with the drugs. Then the plasma was separated by centrifugation at 4000× *g* for 15 min at 4 °C at once and stored frozen at −80 °C until analysis. All the animal handling procedures during the experiments were according to standard operating procedures approved by the institutional animal care and use committee at Northeast Forestry University.

### 3.6. Statistical Analysis

The data were presented as mean ± SD, and the results were obtained from three independent experiments. Statistical analysis was performed with statistics software (SPSS version 17.0). Differences among means were assessed using both one-way analysis of variance (ANOVA) and Tukey’s test. A significance level of *p* < 0.05 was considered to indicate a statistically significant difference.

A non-compartmental pharmacokinetic analysis using the KineticaTM software package (version 5.0, Thermo Fisher Scientific Inc., Waltham, MA, USA) was carried out to calculate the pharmacokinetic parameters including the maximum plasma concentration (C_max_), the time-to-maximum concentration (T_max_), the area under the plasma concentration-time curve from zero to 48 h (AUC_0−48 h_), the area under the plasma concentration-time curve from time zero to infinity (AUC_0−∞_), the elimination half-life time (T_1/2_), and the mean residence time (MRT). The oral bioavailability (F) is measured by comparing AUC_0−∞_ values after i.g. and i.v. administration according to the following equation [19,28]:F = (AUC_i_._g_./Dose_i_._g_.)/(AUC_i_._v_./Dose_i_._v_.).

## 4. Conclusions

To enhance the solubility and bioavailability of BA/BE, we semi-synthesized SBE, which contains an ester bond along with a hydroxyl and carboxylic group similar to BA, by introducing a succinyl group at the C-28 site of BE. Compared to BA, SBE showed significantly higher solubility in most of the tested solvents including water, petroleum ether, acetonitrile, *n*-butanol and methanol. Notably, it exhibited a maximum solubility of 7.19 ± 0.66 g/L in *n*-butanol. Furthermore, our findings revealed that SBE potentially induces cell apoptosis through a mitochondria-dependent pathway, similar to BA. This was evidenced by the increased expression of Bad, as well as an elevated Bad/Bcl-xL ratio and activation of caspase 9. Both SBE and BA exhibited better anti-tumor activity in vitro and in vivo anti-tumor activities, and SBE demonstrated a better potential compared to BA. Further studies will focus on investigating the apoptosis mechanism and further exploring the anti-tumor activities of SBE.

## Data Availability

Not applicable.

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
