# Peer review of "A Novel Betulinic Acid Analogue: Synthesis, Solubility, Antitumor Activity and Pharmacokinetic Study in Rats"

_molecules, 2023, doi:10.3390/molecules28155715_

Round 1

Reviewer 1 Report

The manuscript “A Novel Betulinic Acid Analogue: Synthesis, Solubility, Antitumor Activity and Pharmacokinetic Study in Rats” falls in the field of structural modifications of betulinic acid aimed at improving its solubility and consequent antitumor activity. It describes the synthesis of a betulinic acid analogue, namely 28-O-succinyl betulin, and the study of its cytotoxic activity in vitro and in vivo. This compound showed better hydrophilicity compared to betulinic acid, with improved pharmacokinetic profile due to the higher solubility in water solution. This results in a higher activity compared to betulinic acid and a reduction in tumor growth in xenograft rat model. The authors claim the cytotoxicity is exerted by increase of the expression of Bad, resulting in an increased ratio of Bad/Bcl-xL (correction of the line 365 is suggested) and activation of caspase 9, following the mitochondrial dependent path of apoptosis.

Few observations came up during the revision:

-          Figure 3 is not clear enough. In particular, Bad bands are not sufficiently visible to point out any evaluation. Since this western blot is necessary to prove the mechanism hypothesis, it is of primary importance that this figure appear clear and well defined.

-          References are not homogeneous in format: the name of the journal must be reported always in the same style.

In my opinion, the manuscript is suitable for publication, once these revisions are implemented.

Author Response

Re: Manuscript molecules-2473720

Dear Professor:

Thank you very much for providing us an opportunity to revise our manuscript. We are grateful. We have revised it carefully according to your kind suggestion and helpful comments. The corrected words and sentences are marked in red in the revised manuscript.

Best wishes,

Jian Zheng

---------------------------------

Dr. Jian Zheng

Key Laboratory of Saline-alkali Vegetation Ecology Restoration, Ministry of Education, College of Life Sciences, Northeast Forestry University, Harbin, China

The manuscript “A Novel Betulinic Acid Analogue: Synthesis, Solubility, Antitumor Activity and Pharmacokinetic Study in Rats” falls in the field of structural modifications of betulinic acid aimed at improving its solubility and consequent antitumor activity. It describes the synthesis of a betulinic acid analogue, namely 28-O-succinyl betulin, and the study of its cytotoxic activity in vitro and in vivo. This compound showed better hydrophilicity compared to betulinic acid, with improved pharmacokinetic profile due to the higher solubility in water solution. This result in a higher activity compared to betulinic acid and a reduction in tumor growth in xenograft rat model. The authors claim the cytotoxicity is exerted by increase of the expression of Bad, resulting in an increased ratio of Bad/Bcl-xL (correction of the line 365 is suggested) and activation of caspase 9, following the mitochondrial dependent path of apoptosis.

In my opinion, the manuscript is suitable for publication, once these revisions are implemented.

Few observations came up during the revision:

-          Figure 3 is not clear enough. In particular, Bad bands are not sufficiently visible to point out any evaluation. Since this western blot is necessary to prove the mechanism hypothesis, it is of primary importance that these figures appear clear and well defined.

Response:

Thanks a lot! We have increased the quality of this image.

-          References are not homogeneous in format: the name of the journal must be reported always in the same style.

Response:

We are very sorry for that, and we have revised the references. 

Reviewer 2 Report

As my Post-Doc research Mentor’s words (Prof. Moira L. Bode from University of the Witwatersrand, Johannesburg, South Africa) “Nature doesn’t need Human  Support.. But Human need Nature’s Support.” Most of the current drugs were related to Natural Pharmacophores. The authors did excellent work in search for the novel anti-tumor drugs through simple modification of naturally occurring Betulinic Acid and Betuline.

The present manuscript is well described the scope and significance of the work done by the authors. The following minor corrections will definitely improve the manuscript for publication in this highly reputed journal “Molecules”.

Line 1: bioavailability is not satisfactory…. As the authors stated in line 44 and up to the present study, BA and BE are widely present in nature.. particularly in specific flowers plants and bark of plants and trees (white birch, ber tree, rosemary..etc). Hence, this line could be modified.

 Line 47: Authors can include this citation at Ref. 8 & 9

 https://doi.org/10.1111/cbdd.14148

 Line 76: According the studies, BA and BE has saturated pentacyclic skeleton which is highly lipophilic itself. Hence, the word “lipo-solubility” is not appropriate to use.

Scheme 1: on the reaction arrow, Pyridine (Base) is missed.

Line 237: The reaction was carried out in excess of pyridine (base) (5mL, 62 mmol, 25x to BE). Hence, the final acid product was in the pyridine salt form. I was wondering, how the authors separated or isolated the compound without acidification the reaction mixture.

Hence, the reaction procedure should be rewrite completely.

Line 243, 245: for NMR data.. MHZ should be in the form of MHz. The synthesized compound has total 34 Carbon and 54 Hydrogen atoms. But in the NMR data, only 10 protons were counted. I think some data is missed by mistake. It would be more authentic to submit the H1, 13C NMR and HRMS spectra in support information.

Line 247: For reporting HRMS, Molecular Formula should be reported for calculated ([M-H].+ C34H53O5 541.3893).

The overall quality of English language is good. But should be rechecked before submitting the revised manuscript

Reviewer 3 Report

I suggest authors to use more methods to evaluate apoptosis effect of SBA compound, such as Annex V/PI, as that will be more accurate. Otherwise, BCLxl seems not changed upon SBA treat. You should provide BCL2, BAX protein level as a supporting evidence.

Check carefully throughput the paper and correct some grammar errors.

Author Response

Re: Manuscript molecules-2473720

Dear Professor:

Thank you very much for providing us an opportunity to revise our manuscript. We are grateful. We have revised it carefully according to your kind suggestion and helpful comments. The corrected words and sentences are marked in red in the revised manuscript.

Best wishes,

Jian Zheng

---------------------------------

Dr. Jian Zheng

Key Laboratory of Saline-alkali Vegetation Ecology Restoration, Ministry of Education, College of Life Sciences, Northeast Forestry University, Harbin, China

I suggest authors to use more methods to evaluate apoptosis effect of SBA compound, such as Annex V/PI, as that will be more accurate. Otherwise, BCLxl seems not changed upon SBA treat. You should provide BCL2, BAX protein level as supporting evidence.

Response:

Numerous studies have shown that BA can induce cell apoptosis by activating the mitochondrial pathway. In the case of SBE, a novel analogue of BA/BE, we hypothesized that the apoptosis pathway induced by SBE may also be linked to the mitochondrial pathway.

Therefore, we examined the expression of two apoptosis-related proteins, Bad (a pro-apoptotic protein) and Bcl-xL (an anti-apoptotic protein), which play crucial roles in the mitochondrial pathway. While there was no significant difference in Bcl-xL levels, we observed a notable up-regulation in the Bad/Bcl-xL ratio, along with the activation of caspase 9, following treatment with SBE. These findings suggested that the mitochondrial pathway might be involved in the apoptotic effects of SBE treatment on HeLa cells.

Reviewer 4 Report

In this manuscript, Liang et. al. describe the synthesis, physical properties, and biological activity of a succinyl derivative of betulin. Betulinic acid (BA) and betulin (BE) have anti-tumor properties but their efficacy is impeded by poor bioavailability. A succinyl derivative (abbreviated SBE) of BE was synthesized by esterification of BE with succinic anhydride. It was hypothesized that the carboxyl group could increase water solubility and the rest of the chain in the succinyl group could improve liposolubility. This in turn could potentially enhance the bioavailability of SBE compared to BA or BE. The solubility of SBE was tested in multiple solvents and compared to the solubility of BA. The in vitro cytotoxicity of SBE was compared to that of BA by testing both in several cancer cell lines. In vivo experiments using a xenograft rat model of Lewis lung carcinoma were performed, and the performance of SBE and BA were compared against a placebo group. A pharmacokinetic study was performed by administering SBE both intravenously and orally, and different pharmacokinetic parameters were estimated based on the plasma concentration profile. Finally, western blot analysis was performed to confirm that the mitochondrial apoptosis pathway was involved in the anti-tumor activity of SBE and BA.

I have the following comments and questions for the authors:

(1.) On page 3 (lines 84 to 87, and Table 1), the authors report that SBE had significantly higher solubility relative to BA in water, petroleum ether, acetonitrile, n-butanol, and methanol, but it had relatively lower solubility in ethanol and ethyl acetate. In Table 1, the results for water, petroleum ether, acetonitrile, n-butanol, and methanol are all shown as statistically significant. But the results for ethanol and ethyl acetate are not shown as statistically significant. If these results are not statistically significant, then please remove the phrase “while it showed relatively lower solubility in ethyl acetate and ethanol than BA did” from lines 86-87, page 3. If these results are statistically significant, then indicate this in Table 1 and also explain what could be the reason(s) for these observations. Specifically, methanol, ethanol, and n-butanol are all alcohols; so why does SBE have lower relative solubility in ethanol when it has higher relative solubility in methanol and butanol? And what could be a potential explanation for the lower solubility in ethyl acetate?

(2.) The captions of Tables 1, 2, and 3 mention that “at least three independent experiments” were used for each experimental condition. Please provide specific details on the actual number of replicates that were used for each experimental condition. Additionally, how was statistical significance determined for the results in these tables? The Statistical Analysis subsection (lines 346 to 350, page 9) within the Materials and Methods section only states that SPSS software was used to perform statistical tests without any details on the actual types of tests that were performed. Please include detailed information on the types of statistical tests that were used for all the different types of experiments included in this manuscript, along with any relevant information on why these specific tests were chosen.

(3.) In section 2.3 (pages 3 and 4), it is mentioned that the effects of SBE and BA were assessed at different concentrations between 0 and 50 µM. But Table 3 only shows the IC50 value for each drug. Please include all the results for the different concentrations (both as a table and as graphs) either in the main text or in supplementary information. 

(4.) Lines 117-118 on page 4 state that “These results indicated that the introduction of succinyl moiety on BE might significantly increase the anti-tumor activity of BA analougs.”. However, the results in Table 3 show that the difference in IC50 values between SBE and BA is only about 2-fold at a maximum (it is less than this for several of the cell lines). So even though the results are statistically significant, the effect size does not seem to be too large (the results for the two drugs are within the same order of magnitude). From a clinical translation perspective, is it really justified to say that the succinyl moiety is having a significant impact on the anti-tumor activity? Do we really expect to see a significant clinical improvement based on these results?

(5.) Lines 127 to 131 on page 4 include quantitative information on tumor size and weight, and this has been presented as mean values along with the standard deviations. The standard deviations seem to be rather large, with some of these numbers almost equal to the mean. It is not clear whether this is because of an outlier or if this is the actual amount of variability seen in the data. To clarify this, please include a table and/or graph with the individual data points that were used to calculate these means and SDs (this can be included either in the main text or in the supplementary information).

(6.) In the caption for Figure 2, please include the number of animals included in each group, and also mention what the error bars represent (standard deviations?).  

(7.) Based on the in vivo rat LLC tumor model results, the authors conclude that SBE is more potent compared to BA and has better anti-tumor activity. They include this conclusion in the Abstract, the Results and Discussion, and the Conclusions section. While the SBE treatment group was statistically significantly different from the vehicle control group in terms of tumor volume and tumor weight, the results for SBE compared to BA were not statistically significant. Because of this, it does not appear to be justified to state that SBE is more potent or effective compared to BA. A more accurate conclusion would be to state that both drugs exhibited anti-tumor activity, but there was no significant difference in their effects within this in vivo model. So the conclusion should be that SBE’s anti-tumor effect is comparable to that of BA’s anti-tumor effect (and not better, which is what the authors seem to have concluded).

(8.) Figure 3 (page 5) shows the Western Blot results that demonstrate the signaling pathways activated by SBE and BA. While both SBE and BA treated cells have higher levels of pro-apoptotic proteins, there is no statistically significant difference between the two drugs (Fig. 3B). This further seems to confirm that SBE’s cell penetration and anti-tumor effect is comparable to that of BA’s (and it is not significantly higher/better). 

(9.) Why was Western Blotting chosen to quantitatively assess mitochondrial apoptosis pathway protein levels? The results from a western blot are semi-quantitative at best. Why was a more quantitative method (like ELISA) not used to perform this analysis?

(10.) Figure 3 (page 5) shows snippets of each western blot. Please include images of the complete western blot in the manuscript (either in the main text or the supplementary information).

(11.) Lines 212-215 (page 6) mention that SBE has a much higher absorption after oral ingestion compared to BE. However, based on the results in Figure 4 and Table 4, it appears that SBE is cleared out from the blood stream in 1 to 2 days. Furthermore, the tumor results in Figure 2 did not show any significant results in tumor size between the SBE and BA treated groups. Based on these observations, does SBE have any clinical advantages over BE/BA? Even if SBE is absorbed much better in the gut, it is not retained in the bloodstream for too long and the rat tumor studies show that it did not shrink the tumors significantly compared to BA. Because of this, would it be more accurate to say that SBE is similar to BE/BA in terms of its clinical benefits?

(12.) Lines 278-280 (page 8) state that cells were seeded in a 96-well plate with 0.1-0.2 mL of culture medium per well. How many cells were seeded in each well? And what was the percentage confluence when the drugs were added?

(13.) The in vivo studies all seem to use 6 animals per treatment group. How was this number chosen? Was a statistical power analysis performed?

(14.) Please expand the Conclusions section (page 9). This currently only has a very big picture overview of the work. Please include more details on the important findings (both quantitative and qualitative), more details on the nuances of the findings (including addressing some of the shortcomings above), any areas that were particularly challenging, and potential areas for future research and improvement.

In this reviewer's opinion, this review-manuscript is not publishable in its current form unless the authors directly address these questions and comments by adding appropriate additional information to the relevant manuscript sections.

Round 2

Reviewer 4 Report

Thank you for your responses to my comments from the previous round of peer review. I have a few follow up questions and comments for this round:

(1.) On page 10 (lines 351 to 353) of the revised manuscript, the authors state “Differences among means were evaluated using one-way analysis of variance (ANOVA) and student’s t-test. A significance level of P < 0.05 was considered to indicate a statistically significant difference.”. In cases where ANOVA was used, what post-hoc test was used to calculate the p-values for the pairwise comparisons? If student’s t-test was used for pairwise comparisons following an ANOVA, this is not appropriate as it can have a high false positive rate. Please use a post-hoc test such as the Tukey’s test that corrects for multiple comparisons and report the new p-values. If student’s t test was not used for pairwise comparison following the ANOVA, then please include information on what post-hoc test was used following the ANOVA.

(2.) In response to my question about using quantitative ELISA as opposed to Western Blots, the authors are somewhat vague in their reply. They mention that an advantage of Western Blots is that they can be used to “detect multiple proteins simultaneously”. Multiplex ELISA kits can be used to detect and quantify multiple proteins simultaneously, and this is not an area of disadvantage for ELISA compared to Westerns. The authors also say that ELISA  “typically requires specific antibodies and commercially available kits for each target protein, which may not have been readily accessible or suitable for the specific proteins being studied in this research.” What antibodies/kits were unavailable and/or unsuitable for the proteins of interest in the current manuscript? Are they not available commercially? Or were they not accessible or not suitable because of other reasons? 

(3.) The authors have not responded to my questions in comment # 11 of the previous round of peer review. In that comment, I had stated that even though SBE has higher absorption levels compared to BE/BA, it is cleared out quickly from the blood stream and there was also no significant difference between the SBE and BA groups in tumor size in in vivo studies. The quick clearance and the in vivo tumor size results indicate that SBE may not have a significant clinical advantage over BE/BA despite its higher absorption, and I had asked for the authors’ perspective on this.  In response to my comment, the authors talk about evidence that SBE is absorbed better than BA (which is something I had already acknowledged), but they do not answer my questions within that comment. I have reproduced my comment below for the authors’ convenience. Please provide a response to the questions within it.

Comment 11 from previous round of peer review: “(11.) Lines 212-215 (page 6) mention that SBE has a much higher absorption after oral ingestion compared to BE. However, based on the results in Figure 4 and Table 4, it appears that SBE is cleared out from the blood stream in 1 to 2 days. Furthermore, the tumor results in Figure 2 did not show any significant results in tumor size between the SBE and BA treated groups. Based on these observations, does SBE have any clinical advantages over BE/BA? Even if SBE is absorbed much better in the gut, it is not retained in the bloodstream for too long and the rat tumor studies show that it did not shrink the tumors significantly compared to BA. Because of this, would it be more accurate to say that SBE is similar to BE/BA in terms of its clinical benefits?”

Author Response

Re: Manuscript molecules-2473720

Dear Professor:

Thank you very much for your comments and providing us an opportunity to revise our manuscript. We are grateful. We have revised it carefully according to your kind suggestion and helpful comments. The corrected words and sentences are marked in red in the revised manuscript.

Best wishes,

Jian Zheng

---------------------------------

Dr. Jian Zheng

Key Laboratory of Saline-alkali Vegetation Ecology Restoration, Ministry of Education, College of Life Sciences, Northeast Forestry University, Harbin, China

Thank you for your responses to my comments from the previous round of peer review. I have a few follow up questions and comments for this round:

(1.) On page 10 (lines 351 to 353) of the revised manuscript, the authors state “Differences among means were evaluated using one-way analysis of variance (ANOVA) and student’s t-test. A significance level of P < 0.05 was considered to indicate a statistically significant difference.” In cases where ANOVA was used, what post-hoc test was used to calculate the p-values for the pairwise comparisons? If student’s t-test was used for pairwise comparisons following an ANOVA, this is not appropriate as it can have a high false positive rate. Please use a post-hoc test such as the Tukey’s test that corrects for multiple comparisons and report the new p-values. If student’s t test was not used for pairwise comparison following the ANOVA, then please include information on what post-hoc test was used following the ANOVA.

Response:

Thank you for your suggestion. We have done the Tukey’s test as the post-hoc test and the calculation results for significant differences are consistent as we calculate before. We revised the context as follow: Differences among means were assessed using both one-way analysis of variance (ANOVA) and Tukey's test.

 (2.) In response to my question about using quantitative ELISA as opposed to Western Blots, the authors are somewhat vague in their reply. They mention that an advantage of Western Blots is that they can be used to “detect multiple proteins simultaneously”. Multiplex ELISA kits can be used to detect and quantify multiple proteins simultaneously and this is not an area of disadvantage for ELISA compared to Westerns. The authors also say that ELISA “typically requires specific antibodies and commercially available kits for each target protein, which may not have been readily accessible or suitable for the specific proteins being studied in this research.” What antibodies/kits were unavailable and/or unsuitable for the proteins of interest in the current manuscript? Are they not available commercially? Or were they not accessible or not suitable because of other reasons? 

Response:

Thank you for your suggestion. In the present study, we aimed to detect the protein levels of apoptosis-related proteins in order to confirm the potential apoptosis pathway induced by SBE. To achieve this, we utilized Western blotting, which proved to be a suitable method for our purpose. This technique also allowed us to obtain relatively quantitative results while addressing the research question in a cost-effective manner.

(3.) The authors have not responded to my questions in comment # 11 of the previous round of peer review. In that comment, I had stated that even though SBE has higher absorption levels compared to BE/BA, it is cleared out quickly from the blood stream and there was also no significant difference between the SBE and BA groups in tumor size in in vivo studies. The quick clearance and the in vivo tumor size results indicate that SBE may not have a significant clinical advantage over BE/BA despite its higher absorption, and I had asked for the authors’ perspective on this. In response to my comment, the authors talk about evidence that SBE is absorbed better than BA (which is something I had already acknowledged), but they do not answer my questions within that comment. I have reproduced my comment below for the authors’ convenience. Please provide a response to the questions within it.

Comment 11 from previous round of peer review: “(11.) Lines 212-215 (page 6) mention that SBE has a much higher absorption after oral ingestion compared to BE. However, based on the results in Figure 4 and Table 4, it appears that SBE is cleared out from the blood stream in 1 to 2 days. Furthermore, the tumor results in Figure 2 did not show any significant results in tumor size between the SBE and BA treated groups. Based on these observations, does SBE have any clinical advantages over BE/BA? Even if SBE is absorbed much better in the gut, it is not retained in the bloodstream for too long and the rat tumor studies show that it did not shrink the tumors significantly compared to BA. Because of this, would it be more accurate to say that SBE is similar to BE/BA in terms of its clinical benefits?”

Response:

Thank you for your suggestion. We revised the statement in the abstract and conclusion from “SBE demonstrated a better potency compared to BA” to “SBE demonstrated a better potential compared to BA”.
